# The Role of Associations in Reducing the Emotional and Financial Impact on Parents Caring for Children with Duchenne Muscular Dystrophy: A Cross-Cultural Study

**DOI:** 10.3390/ijerph191912334

**Published:** 2022-09-28

**Authors:** Alicia Aurora Rodríguez, Imanol Amayra, Juan Francisco López-Paz, Oscar Martínez, Maitane García, Mónika Salgueiro, Mohammad Al-Rashaida, Paula María Luna, Paula Pérez-Nuñez, Nicole Passi, Irune García, Javiera Ortega

**Affiliations:** 1Neuro-e-Motion Research Team, Faculty of Health Sciences, University of Deusto, Av. Universidades, 24, 48007 Bilbao, Spain; 2Department of Clinical and Health Psychology, and Research Methodology, Faculty of Psychology, University of the Basque Country, Tolosa Hiribidea, 70, 20018 Donostia, Spain; 3Department of Education, College of Arts and Sciences, Dubai Campus, Abu Dhabi University, Abu Dhabi 59911, United Arab Emirates; 4Centro Investigaciones de Psicología y Psicopedagogía [CIPP], Facultad de Psicología y Psicopedagogía, Pontificia Universidad Católica Argentina, Buenos Aires 1107, Argentina

**Keywords:** Duchenne muscular dystrophy, caregivers, economic costs, psychological health, quality of life, non-governmental organizations, Mexico, Spain

## Abstract

Caregivers’ emotions and finances are affected by the deterioration of functional capacity of patients with Duchenne muscular dystrophy (DMD), both in Mexico and Spain. Patient associations may reduce this impact on caregivers. This study aims to study the role of two models of associations, inspired by two different cultural models, in how the services they provide can help decrease the emotional and financial impact on the caregivers of children with DMD. The sample consisted of 34 caregivers from Mexico and 40 from Spain recruited from Spanish hospitals and rare disease organizations in Spain and Mexico. The instruments used consisted of a sociodemographic and socioeconomic questionnaire, the CarerQol-7D, the PHQ-15, the Zarit Caregiver’s Burden Scale and the SWLS. The results showed that caregivers in Mexico are in better physical and psychological health than caregivers in Spain. They also receive more subsidies than those in Spain. Caregivers in Mexico have a greater well-being and are less affected by the economic impact of the disease due to the associations’ day-to-day work and the fact that they generate a network of health services that they make available to the patient free of charge. These differences may also be attributable to cultural issues and to the fact that Mexico has a deeply established culture of support.

## 1. Introduction

Over the years, the interest in and awareness of rare diseases (RDs) have grown in both the state and the society. RDs are chronic, low-prevalence conditions, characterized by diagnostic delay, uncertain clinical course and lack of curative treatments [1]. Some cross-cultural studies have attempted to analyze the economic and emotional impact of a chronic illness on patients’ lives [2,3] and on the caregivers of children with RDs [4,5] and how the social model of each country can influence this impact. However, to our knowledge, none of these studies considered the different cultural benefits provided by associations from two different countries to RDs patients’ caregivers. The structure and aims of the services provided by the associations are determined by public health policies.

When comparing the public policies in Latin America and Europe, public policies related to RDs were developed at a relatively late stage in Latin America [1]. As a result, RD patients are only partially cared for. In policy terms, these diseases have been associated with catastrophic diseases, high-cost diseases, chronic non-communicable diseases and the disabled population. However, RDs have not been specifically addressed by the authorities. Six countries, including Mexico, have introduced legislation on RDs in Latin America [6]. Nonetheless, the national plans, programs and strategies that deal with the needs of this group are missing [7]. Consequently, patient associations are the ones that have taken the responsibility to look out for the RDs population’s well-being [6]. Furthermore, RDs have become more visible and gathered more interest thanks to patient federations and organizations [6].

The development of policies aimed at safeguarding the rights of RDs in the EU has followed a different course, starting earlier and consolidating between the years 2000 and 2004 [8]. In Spain, a national strategy was presented back in 2009 in order to adapt healthcare actions to the needs of patients and to better monitor their implementation [9]. The monitoring showed that healthcare and social systems in Spain do not adequately meet the needs of RD patients, which is similar to the situation in Mexico [10,11].

Due to the lack of public funds, there is a decrease in the income of patients and their families [12,13,14]. It is estimated that they spend a large part of their income on medicines, travel, home adaptations, complementary therapies and unique food products [15]. This can lead to problems in the family budget, which is further aggravated if any of the family members have to reduce their working hours or quit their jobs due to illness [16].

In Mexico, non-governmental organizations provide comprehensive and multidisciplinary care support for RD patients and their families, thereby reducing this economic imbalance [17]. They offer guidance, disseminate information, raise awareness, sensitize the population and provide social education about RDs [18].

The situation is similar in Spain [19]. Support organizations for people with RDs are essential for their visibility and are considered a public health priority in Europe [20]. In addition, EUROPLAN [21] leads actions to promote research in this area and encourage collecting data on these RDs. The purpose is to assess the actual situation in Spain and other countries regarding these conditions.

Support organizations for people with RDs led to the establishment of a network, which, among other goals, aims to spread knowledge among the public to improve the current understanding of these diseases. They have also played a key role in improving care and research on RDs, as they proposed a joint strategy and a series of actions to ensure that the needs of patients and their families were met [22].

Some associations in Spain and Mexico give visibility and support to a group of rare diseases called neuromuscular disorders (NMDs). One of these is Duchenne muscular dystrophy (DMD), an X-linked recessive hereditary disorder, where progressive muscle degeneration leads to the loss of the ability to walk independently by the age of 13 [23,24,25,26]. The prevalence of DMD is less than 10 cases per 100,000 males and seems to be the same between regions, and DMD in females is very rare (<1 per million) and is limited to case reports of individuals with the Turner syndrome, a translocation involving DMD or those with bi-allelic DMD mutations [27].

DMD entails a high degree of disability and dependency. Patients with DMD need constant care and supervision by informal caregivers–parents or other first-degree relatives. Given that this type of care requires specific expertise, takes a long time and is associated with additional financial costs, its impact on the caregiver’s life is noticeable [28].

Many caregivers of children with DMD quit their jobs or reduce their working hours to care for them. For those who continue to work, the productivity declines due to frequent absences from work. In addition to the economic impact resulting from job loss, the affected families bear substantial costs associated with insurance premiums and medical care co-payments [29,30].

Additionally, the caregivers’ emotions, family life and social life are affected due to the deterioration of patients’ functional capacity [31]. The child’s disability causes changes in the caregiver’s daily life activities. Caring for a person with DMD is also very stressful and can impact the caregiver’s physical and psychological well-being [30]. Studies have shown that caregivers suffer from high demands, stress, overload, distress and lower quality of life (QoL) [32].

The caregivers of people with these pathologies suffer from a significant burden, causing an impediment to their social life [33]. Furthermore, caring for a dependent person not only affects psychological well-being but can also reduce the caregiver’s physical health [34]. In addition, their perceived satisfaction with life, and thus, their subjective well-being, is also worse than that of parents of children without any illness [35].

Due to the impact of DMD on the caregivers’ life, the associations work to help caregivers overcome the challenges they face [36]. However, there are no previous studies that analyze the positive impact of associations on the lives of parents with children with DMD.

It is particularly worth analyzing two models of health care to better understand how associations play an important role. In Spain, health care consists of public health care, so patients and caregivers do not have to bear these costs, and the associations take care of other aspects, such as social support. However, this public healthcare is not simultaneous care, i.e., parents are generally not scheduled for all health examinations on the same day and often have to take over the costs of private specialists because they are not satisfied with the care received. Contrariwise, in Mexico, although healthcare is private, the associations cover the cost of care and provide the patient with a team of health professionals who all see the patient on the same day. Therefore, care is comprehensive. These cross-cultural differences are not only related to the type of care but also to other psychosocial variables. The type of culture, tradition or social structure is also reflected in patient organizations [37].

So far, no studies have analyzed collectively and comparatively the impact of different models of associations in terms of the services provided on caregivers’ well-being. Regarding the last point, there have also been no studies specifically focused on the role of associations in different countries in reducing the impact of the disease on caregiver well-being in terms of financial costs, QoL, life satisfaction, burden and somatic symptoms. The originality of the present study resides in the analysis of two different models of associations, the differences between which are based on culture and the health care system.

Because of all the above, this study aims to evaluate the role of two models of associations, inspired by two different cultural models, in reducing the emotional and financial impact on the caregivers of children with DMD. This study is the first to our knowledge to conduct a comprehensive analysis of the interaction between the mentioned variables. Specifically, this study describes the financial costs arising from illness, burden, life satisfaction, QoL and somatic symptomatology in caregivers of children with DMD at a cross-cultural level in Mexico and Spain.

## 2. Materials and Methods

### 2.1. Participants

The sample consisted of 74 caregivers of children with DMD from Spain and Mexico. Forty caregivers from Spain were recruited from various DMD organizations (ASEM and BENE), from the Hospital de Basurto and the Hospital Universitario de Cruces. Thirty-four caregivers from Mexico were recruited from the Coalición Latinoamericana de Duchenne-Becker. All participants were members of DMD organizations.

The inclusion criteria were the following: (a) To be an informal caregiver (parent) of a child diagnosed with DMD; (b) be over 18 years of age; (c) to be willing to sign the informed consent document before participating in the study; and (d) to be a resident in Spain or Mexico and have Spanish as one of the primary languages of communication.

Exclusion criteria: (a) informal caregiver of a child with any other diagnosis not secondary to the diagnosis of DMD; (b) any other psychological or psychiatric diagnosis not secondary to DMD; (c) uncompensated sensory deficits that prevent the administration of the evaluation protocol; and (d) illiteracy.

### 2.2. Instruments

#### 2.2.1. Measures

##### Sociodemographic Data

The first instrument was a 17-item ad hoc questionnaire, which collected the participants’ sociodemographic data (e.g., sex, age, academic level, type of employment and marital status).

##### Economic Data

A 248-item semi-structured questionnaire was used to assess the financial costs associated with care. The questionnaire includes questions on assistive technology that the family had to buy, the money spent on activities to improve the health of both the child and the caregiver, the time spent engaging in these activities and the loss of work productivity. The subsidies paid by the state and the funding provided by the associations to cover the expenses are also assessed. For example, a number of questions comprising this questionnaire ask about the costs of medical care, physiotherapy, psychological care, purchase of wheelchairs, splints, etc. This questionnaire was developed by Rodríguez et al. [31] and was supported by other studies conducted with people with NMD [38,39,40,41,42].

##### Somatic Symptoms

The PHQ-15 [43] was used to assess the somatic symptoms associated with care. This questionnaire consists of 15 items referring to 15 possible physical problems that caregivers may have had in the previous four weeks. The total PHQ-15 score ranges from 0 to 30, and scores of >5, >10, >15 represent mild, moderate and severe levels of somatization. The possible response options to each one of the 15 items are: “not bothered”, or absence of a physical problem (0 points), “a little”, or presence of a problem (1 point), or “a lot”, or significant presence of a problem (2 points) [43]. Studies, such as that by Montalbán et al., [44] have demonstrated high internal consistency (Cronbach’s alpha 0.78). The Mexican version obtained a Cronbach’s Alpha of 0.77 [45]. The Cronbach’s alpha coefficient in this study was 0.88.

##### Caregiver Burden Scale

The Zarit Caregiver Burden Scale [46] was used to assess the participants’ burden due to the caring experience. The scale measures the workload of caregivers of dependent people and determines the burden of the caregiver’s experiences as an overall score. It is a one-dimensional scale that consists of 22 items, measured on a four-point Likert-type response scale. It has been shown to have a good internal consistency in Spain (Cronbach’s alpha 0.91) [47] and Mexico (Cronbach’s alpha 0.84) [48]. In the current study, Cronbach’s alpha coefficient was 0.90.

##### Satisfaction with Life

The Satisfaction with Life Scale (SWLS) is a unidimensional scale employed to assess the caregiver’s satisfaction with life [49]. The SWLS measures the individual’s level of satisfaction with life at that moment. It consists of five questions, where each question is rated on a 7-point scale (1 = do not agree at all with the item, 7 = strongly agree with the item). The possible range of this scale is from 1 to 7 per question. However, the Spanish version is rated on a 5-point scale. Life satisfaction refers to a subjective cognitive process in which people judge their overall satisfaction with their current situation concerning self-defined standards or expectations of what they would like their life to be [50]. Cronbach’s alpha was 0.89 [51]. In the Spanish version, the scale was shown to have high internal consistency, with Cronbach’s alpha coefficients ranging from 0.79 to 0.89 [52]. The Mexican version obtained a Cronbach’s alpha of 0.83 [53]. A Cronbach’s alpha coefficient of 0.88 was obtained in the study.

##### Quality of Life

QoL was assessed using the CarerQol [54]. This questionnaire measures care-related QoL and consists of 7 items. The intraclass correlation coefficients (ICCs) of the CarerQol7D show values between 0.55 and 0.94 [54]. The CarerQoL includes two components. The first component is the CarerQol-VAS, which measures well-being in terms of happiness using the visual analog scale (VAS), with the endpoints being “completely unhappy” and “completely happy”. The second component is the CarerQoL-7D, which measures subjective burden through seven items that evaluate the following dimensions: fulfillment, relational problems, mental health, financial problems, daily life activities, external support and physical health [55]. The utility rates for the CarerQol were developed to calculate a CarerQol-7D utility score from the responses to the seven dimensions, ranging between 0 (“worst imaginable caregiving situation”) and 100 (“best imaginable caregiving situation”), for which discrete choice experiments were used [56]. The CarerQol-7D had a Cronbach’s alpha of 0.641 [57] and a Cronbach’s alpha of 0.62 in another study [58]. The present study obtained a Cronbach’s alpha coefficient of 0.63.

### 2.3. Procedure

The sample was recruited from different support associations of patients with DMD and two hospitals in Bizkaia. Those interested in participating in the study were informed about the evaluation process. The data were collected using a self-administered protocol via the “Qualtrics” virtual platform, accessed through a personal link. All caregivers agreed to participate in the study by giving consent before completing the survey. They were provided with a telephone number to contact the researcher if they wished to ask any questions while completing the survey. The duration of the protocol was approximately one hour. The Responsible Ethics Commission approved the research (Ref: ETK-39/18-19), which was conducted in accordance with the Declaration of Helsinki.

### 2.4. Statistical Analysis

Descriptive statistics were used to describe the participants. The continuous variables were described by mean and standard deviation, and the categorical variables by frequency and percentage. Regarding the socio-economic variables, the mean scores obtained from the structured items were calculated. Similarly, the mean scores of the PHQ-15, Zarit and SWLS global scores were calculated. For the CarerQoL-Tariff, the syntax provided by the authors of the instrument was applied. The financial costs in Spain were expressed in EUR, while in Mexico, they were described in MXN. All expenses were then converted to EUR to facilitate comparison. The Kolmogorov–Smirnov test was used to test for normality of the outcome variables. Most variables did not have a normal distribution because the coefficient (K-S) was significant (*p* > 0.05). Data were analyzed using the Mann–Whitney U-test and Chi-square test. A *p*-value below 0.05 was defined as statistically significant. IBM SPSS Statistics 26.0 was used for all analyses.

## 3. Results

Forty caregivers of children with DMD from Spain and thirty-four caregivers of children with DMD from Mexico participated in the study. Table 1 shows the data related to the sampling distribution according to the caregiver’s sex, marital status, academic level and employment status. Regarding the sociodemographic profile of both groups of caregivers, there were a more significant number of male caregivers in the Spanish sample and a greater number of women in the Mexican sample (χ^2^ (1) = 9.82, *p* = 0.02). As far as the marital status was concerned, married people predominated in the Spanish sample, while in Mexico, not having a partner or not being married prevailed (χ^2^ (5) = 5.90, *p* = 0.023). In terms of employment, most Spanish caregivers were employees, while the Mexican caregivers were engaged in housework (χ^2^ (8) = 17.56, *p* = 0.025).

The mean age of the Spanish caregivers was 46.15 ± 8.14, and the mean age of the Mexican caregivers was 41.18 ± 10.14. The average years of education of the caregivers were 14.09 ± 7.02 (Spain) and 10.84 ± 4.57 (Mexico); the average number of children was 2.15 ± 2.53 (Spain) and 2.53 ± 0.86 (Mexico), and the average age of the child with NMD was 14.58 ± 5.38 (Spain) and 13.15 ± 4.63 (Mexico). Statistically significant differences were observed between Mexico and Spain in caregiver age (U = 398.50, *p* = 0.002), years of education (U = 384, *p* = 0.027) and the number of children (U = 448, *p* = 0.006).

First, the following monthly expenses incurred by caregivers in both Mexico and Spain were calculated: fees expended on health professionals for the child; costs expended on health professionals for the caregivers; annual expenses on assistive technology for the child; and the amount of money subsidized for each expenditure. The expenses and subsidies are presented in EUR for easy comparison.

As household income largely influences the impact of household expenses, the annual salary in both countries was also assessed. In Spain, the average yearly salary of a family unit was 38,197.16 ± 43,350.57. In Mexico, it was 8272.47 ± 20,762.94. Given the salary difference between the two countries, the percentage of household income spent on these expenses was assessed to see the real impact on the family. The following formula was used to calculate the percentage of family income allocated to each payment: (Annual service cost—annual subsidies granted)/Annual household income. The percentages are shown in Table 2.

Descriptive analyses were carried out on the psychological variables of burden, life satisfaction, somatic symptoms and QoL. An analysis was also carried out to study any statistically significant differences in these scores between Mexico and Spain. For somatic symptoms, the scores for Spain were 12.42 ± 7.22, a medium degree of severity, and 7.65 ± 5.38 for Mexico, a mild degree of severity. These differences in somatic symptoms were found to be statistically significant (U = 244, *p* = 0.011). In terms of caregiver’s burden, the scores were 35.06 ± 15.46 for Spain, corresponding to “mild to moderate” severity, and 25.07 ± 12.44 for Mexico, corresponding to “mild to moderate “overload. The differences in caregiver’s burden scores were statistically significant (U = 287.5, *p* = 0.028). Regarding life satisfaction, caregivers scored 14.35 ± 5.66 (Spain) and 18.33 ± 3.69 (Mexico). The Spanish adaptation of this scale has a 5-point Likert scale; therefore, the minimum score that could be obtained was 5, and the maximum score was 25. The other international versions have a 7-point Likert scale [59]. The differences were statistically significant in life satisfaction (U = 214, *p* = 0.007). For QoL, scores of 43.59 ± 26.58 (Spain) and 46.99 ± 26.99 (Mexico) were obtained.

Regarding the number of hours per week spent by caregivers on supporting their child, the average number of hours in Spain was 44.62 ± 37.75, whereas the average number of hours in Mexico was 35.29 ± 49.45.

There were statistically significant differences between Mexico and Spain regarding the expenses associated with health professionals who provided services for the child and the caregiver and ones associated with assistive technology. These differences were found in expenses related to assistive technology (U = 444, *p* = 0.010), in subsidies for health professionals to provide services for the child (U = 440, *p* = 0.010), in subsidies for assistive technology (U = 361, *p* = 0.000) and in subsidies for health professionals to provide services for the caregiver (U = 600, *p* = 0.027). All the other analyses indicated no statistically significant differences (Table 3).

## 4. Discussion

Patient associations play a protective role in families, reducing the disease’s psychological and economic impact. They have been shown to positively affect research development, government regulations and clinical trials [35]. For this reason, this study examined the role of organizations in reducing the emotional and economic impact of illness on caregivers, analyzing the financial expenses incurred by caregivers of people with DMD in Mexico and Spain and the physical and psychological consequences of providing care. Considering these variables, the role of associations in the care of parents of children with DMD was analyzed.

While the socioeconomic situation in each of these countries differs in many aspects, it was found that the economic impact of DMD on these families was high and significant in both countries. Health and social systems do not adequately address the needs of patient with these conditions. Families in both countries spend a large part of their income on health professionals for both the child and the caregiver. However, the DMD patient support associations play an essential role in reducing these families’ economic and psychological impact.

To start with, the highest annual expenses incurred by these families were related to assistive technology for the care of their child, followed by costs associated with health professionals to treat the child and, lastly, expenses related to professionals providing services for the caregiver. Not surprisingly, in both Spain and Mexico, the highest costs were linked to purchasing assistive technology to improve mobility, as the average age of the patients in this study was between 13 and 14 years old. It is from this age group onwards that the degree of disability increases. Similarly, another study that focused on family members of people with rare NMDs found that assistive technology usually involves increased costs for the family [12]. These expenditures on technical aids were significantly different between the two countries. Mexican caregivers were found to spend less money than Spanish caregivers on technical aids, this being the most prominent expenditure observed in the present study. One reason for this difference is the fundamental work of the associations in providing the patient with technical aids, such as wheelchairs or other orthopedic aids.

On top of that, these data become even more substantial when considering the percentage of household income allocated to costs associated with the disease. It should be noted that the minimum income per family unit is different in Mexico and Spain. Therefore, it is vital to consider each family’s minimum income and the costs and subsidies received. The highest household income percentages in Spain and Mexico were also found in assistive technology costs. However, the rates were also remarkable regarding the costs associated with professionals supporting the child and the caregiver. For Mexico, these results were consistent with a previous study that estimated that the families of people with DMD spend much of their income on care [60]. In the case of Mexico, it is the association that reduces the impact.

In terms of the number of hours spent providing care per week, it was higher in Mexico and Spain than in previous studies focused on the caregivers of people with schizophrenia [61], caregivers of older people [62] and caregivers of people with dementia [63]. However, there were no significant differences between the two countries.

Moreover, there were differences between Spain and Mexico in terms of subsidies. The results showed that the subsidies directed at health services are higher in Spain than in Mexico. This difference may be due to the fact that, in Spain, the health system is public, and therefore, there is a wider range of public services available to the patient, free of charge. However, when analyzing the overall impact of these expenses and subsidies on the caregiver’s life, considering their salary, there were no significant differences. The same phenomenon was observed when studying the subsidies for technical aids. At the first glance, technical aids subsidies are higher in Spain than in Mexico, but when considering all the financial factors, the economic impact is higher in Spain. Therefore, when economic impact studies are conducted, it is relevant to consider the salary of the person, the subsidies received and the emotional impact.

On the other hand, it can be observed that caregivers of children with DMD receive more subsidies for the caregiver in Mexico than in Spain. It is important to note that the funding sources are different in the two countries; in Mexico, they come mainly from patient support associations [14], while in Spain, they originate from both the state and the associations for people with DMD [64].

The interventions in Mexico are multidisciplinary and come from the associations themselves. The associations provide patients and families with the services of professionals (neurologists, pulmonologists, cardiologists, etc.) necessary to provide comprehensive care for their needs. As mentioned above, this cross-cultural analysis is based on two different models. In the Mexican case, each patient is cared for simultaneously by other professionals, although the patient organizations do not provide this clinical coordination in Spain. In Spain, there is a very fragmented model, in which care is public and free but not comprehensive, so care will not be simultaneous. In this case, the associations demand that patients receive complete and simultaneous care. Therefore, multidisciplinary treatment and care should be a positive element to consider when establishing the action plans to reduce the impact of the disease in this group.

Organizations struggle every day to improve their QoL and reduce the impact of the disease. They are actively involved in fundraising for research in the hope of finding a cure for DMD. They also encourage the creation of patient registers to facilitate the recruitment of participants for clinical trials [65]. Many associations provide the medical services that the child needs, which considerably reduces the costs associated with the disease [66].

In addition to the economic costs, the disease has a significant emotional impact on families. Caregivers in Spain had scores consistent with medium-severity somatic symptoms and mild to moderate burden levels. In contrast, caregivers in Mexico had scores consistent with mild symptoms and burden levels. The presence of somatization in caregivers is common in the literature [67,68]. This is consistent with the fact that caring for a person with dependency not only affects psychological well-being but can also have a negative impact on the caregiver’s own health [35]. The results obtained in somatic symptomatology were higher in the case of Spain and lower in the case of Mexico compared to other studies with patients with anxiety and depression [44], with caregivers of people with Alzheimer’s disease [69] and with caregivers of patients with RDs [70]. The same occurred in the overload scores. The results obtained were higher for the Spain sample and lower for the Mexico sample than the results observed in caregivers of patients with RD, specifically with spinal muscular atrophy [71] and with Wolf–Hirschhorn [72]. In the case of studies carried out specifically in Mexico and with a normative population, we found that the score obtained in the present sample is higher in both cases [73].

The presence of emotional disturbances in caregivers of dependent persons has been widely studied and corroborated by different studies [74,75,76]. The patient’s dependency in their daily life causes progressive physical and mental deterioration in the caregiver, which brings along depressive symptoms and puts their psychological health at risk [77]. Similar results were found in the present study. Other authors have described that caregivers of children with NMD typically report high levels of stress related to the patient’s demands, acceptability, social isolation, [78] and low QoL [79]. Other studies have shown that these caregivers suffer from high levels of anxiety and depression because of their caregiving responsibility [79,80].

A reduction in life satisfaction can also be observed within the physical and psychological impact. It is noted that caregivers feel that they have not achieved what they wanted in life and that their life circumstances are not favorable. The mean life satisfaction in the study was lower than in other studies conducted on a “normative” population [81,82,83]. However, the mean life satisfaction in Mexico was similar to that found in a previous study with a Mexican population [53].

Concerning the scores obtained on the CarerQoL scale, Mexico and Spain demonstrated lower QoL levels than other studies related to informal caregivers [56,57,84,85].

Considering a cross-cultural comparison, Spanish caregivers were found to have more severe somatic symptoms, higher burden levels and lower life satisfaction than Mexican caregivers. These results must be contextualized considering the culture of each country, bearing in mind that caregivers from Mexico were exclusively women. For Mexican women, motherhood is a social requirement that gives meaning to their lives, as it is a constitutive element of “feminine identity”. Women have always been required to take care of the children’s upbringing, care for the sick and provide affection and support as mothers/wives. Therefore, women adapt their lifestyles to prioritize their family and parenting roles as part of their identity [86]. Because of this, the psychological and physical stress related to their child’s illness is likely to be attenuated by integrating it into their responsibility as mothers. Another reason that may explain these differences is the comprehensive care mothers receive, which was pointed out throughout the study. The reassurance of mothers that their children will be comprehensively cared for by several specialists may be influencing the lower emotional impact on Mexican mothers.

Furthermore, as mentioned above, Mexico is characterized by a strong culture of associations dedicated to supporting rare diseases. Social support is a vital resource in coping with an illness of a family member and a variable that may reduce the impact of the condition [87]. The decisive role played by these associations in supporting the families may minimize the emotional consequences of a child’s disease.

Overall, the present study demonstrates differences in the status of caregivers belonging to associations in Mexico and Spain. It is observed that caregivers of children with DMD in Mexico have a greater well-being and are less affected by the economic impact of the disease, which is attributable to the work carried out by the associations on a daily basis and to the fact that they generate a network of social support and health services that they make available to the patient free of charge. On the other hand, these associations assume competencies that should be national governments’ responsibilities in the first place. These differences may also be attributable to cultural issues and to the fact that Mexico has a deeply rooted culture of support. This study shows that an optimal support network and comprehensive multidisciplinary medical and health care would improve the lives of both patients and their families, reducing the negative impact of the disease on the family. The fact that a child with DMD can receive all treatments on the same day also increases the parents’ tranquility.

The limitations of this study are related to the length and duration of the protocol used. As discussed above, difficulties were also encountered in drawing comparisons between adaptations of the Satisfaction with Life Scale at the international level. Additionally, the lack of studies on the physical and psychological states of the caregivers of children with DMD also makes international comparisons difficult. Another limitation was the type of sampling used, which was by convenience, as these are rare diseases. On the other hand, another limitation found in the present study was the measurement of QoL obtained through the CarerQoL, which is global and unidimensional, not allowing us to exhaustively assess which dimensions of QoL are most affected. Another limitation we found is that the present study only includes fathers and mothers, although informal caregivers can be siblings, grandparents, aunts and uncles, etc.

Future research could seek to generate a unified protocol to assess the physical and psychological state of caregivers of patients with neuromuscular disorders in order to plan interventions that consider this group’s specific needs. Furthermore, there is a need for instruments that explore in greater depth the dimensions of QoL that are specific to this population. So far, other generic subscales for pediatric diseases, such as the PedsQL FIM, have not been validated with a Spanish sample. It would also be interesting to conduct a comparative study with other Latin American countries. Finally, a longitudinal study would be appropriate to assess the positive impact that organizations have on families over time.

The practical implications of the present study include that a multidisciplinary care model should be considered as a positive element. The results obtained demonstrated how this care positively affects patients and families, so it should be considered in future studies and when developing health intervention plans. Likewise, the fact that Mexican caregivers have a higher overall well-being suggests that having a cultural value of social support is fundamental for the well-being of this population, as well as multidisciplinary and quality medical care. On the other hand, gender variables should be considered in future research.

## 5. Conclusions

In general, the participants’ caregivers in Mexico have better physical and psychological health than those in Spain. In addition, the economic impact of the disease is considerably reduced in Mexico compared to Spain, thanks to the role of patient associations. The associations ensure that each patient is treated simultaneously by different professionals in the Mexican case, although the patient organizations do not provide this clinical coordination in Spain. In this case, the associations demand that patients receive comprehensive and simultaneous care. In Spain, there is a fragmented model in which care is public and accessible but not complete. Therefore, many parents assume the costs of private specialists because they are not satisfied with the care they receive. Integrated, comprehensive and multidisciplinary care, as well the development of social support programs for caregivers, would help improve the lives of patients and their families. Since healthcare systems often fail to address all of the family’s needs, DMD patient support organizations play a crucial role in reducing the economic and psychological impact of the disease on these families. At the individual and social level, cultural values such as social support should be promoted for the well-being of this group, as well as governmental measures to develop a range of health services that address the needs of these patients.

## Figures and Tables

**Table 1 ijerph-19-12334-t001:** Distribution of sociodemographic and medical variables.

Variable		Spain (*n* = 40)	Mexico (*n* = 34)
Gender	Male	10 (25%)	0 (0%)
	Female	30 (75%)	34 (100%)
Marital status	Married	32 (80%)	17 (48.57%)
	Living as a couple	3 (7.5%)	6 (17.64%)
	Divorced	2 (5%)	7 (20.58%)
	Separated	3 (7.5%)	1 (2.94%)
	Single	0 (0%)	1 (2.94%)
	Widow/er	0 (0%)	2 (5.88%)
Occupation	Employee	16 (40%)	7 (20.58%)
	Self-employed	3 (7.5%)	2 (5.88%)
	Unpaid work	2 (5%)	0 (0%)
	Unemployed (for health reasons)	1 (2.5%)	1 (2.94%)
	Unemployed (for other reasons)	6 (15%)	2 (5.88%)
	Retired	1 (2.5%)	3 (8.82%)
	Housework	5 (12.5%)	17 (50%)
	Student	5 (12.5%)	1 (2.94%)
	Disabled	1 (2.5%)	1 (2.94%)

**Table 2 ijerph-19-12334-t002:** Percentage of annual income related to caregivers’ health expenses (*n* = 74).

	Spain Annual Percentage (%) M ± SD	Mexico Annual Percentage (%) M ± SD
Expenses associated with services provided by health professionals for the child	4.61 ± 9.61	5.89 ± 13.73
Expenses related to assistive technology	8.68 ± 17.23	7.42 ± 18.02
Expenses associated with services provided by health professionals for the caregiver	0.73 ± 2.05	3.45 ± 11.20

**Table 3 ijerph-19-12334-t003:** Mann–Whitney U-test analysis of clinical variables and nationality (*n* = 74).

Clinical Variables	Sociodemographic Variables	*n*	Mid-Range	Average	U	*p*	R
Expenses associated with health professionals for the child	Spain Mexico	40 34	40.20 34.32	1384.10 151.58	572	0.191	0.320
*Expenses associated with assistive technology*	Spain Mexico	40 34	43.40 30.56	44734.12 3487.66	444	0.010	0.300
Expenses associated with health professionals for the caregiver	Spain Mexico	40 34	37.54 37.46	213.30 75.98	678.5	0.984	0.140
*Subsidies for expenses associated with health professionals for the child*	Spain Mexico	36 34	30.72 40.56	161.36 49.52	440	0.010	0.130
*Subsidies for assistive technology*	Spain Mexico	40 34	45.48 28.12	2884.75 249.00	361	0.000	0.380
*Subsidies for expenses associated with health professionals for the caregiver*	Spain Mexico	40 34	35.50 39.85	0.00 22.05	600	0.027	-
Percentage of family income spent on health professionals for the child	Spain Mexico	32 28	32.91 27.75	0.04 0.05	371	0.192	−0.05
Percentage of household income spent on assistive technology	Spain Mexico	35 27	33.63 28.74	0.08 0.07	398	0.281	0.122
Percentage of family income spent on health professionals for the caregiver	Spain Mexico	39 31	34.49 36.77	0.00 0.03	565	0.565	−0.173
Hours spent on care	Spain Mexico	40 34	41.00 33.38	44.62 35.29	540	0.123	0.106
*Somatic symptoms*	Spain Mexico	31 26	34.13 22.88	12.42 7.65	244	0.011	0.347
*Satisfaction with life*	Spain Mexico	31 24	22.90 34.58	14.35 18.33	214	0.007	−0.379
*Caregiver’s burden*	Spain Mexico	32 27	34.52 24.65	35.06 25.07	287.5	0.028	0.332
Quality of life	Spain Mexico	40 34	36.19 39.04	43.59 46.99	627.5	0.568	0.063

## Data Availability

The datasets generated and/or analyzed during the current study are not publicly available because they belong to the University of Deusto, but they are available from the corresponding author (Alicia Aurora Rodríguez) on reasonable request.

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
