# Peer review of "The Role of Associations in Reducing the Emotional and Financial Impact on Parents Caring for Children with Duchenne Muscular Dystrophy: A Cross-Cultural Study"

_ijerph, 2022, doi:10.3390/ijerph191912334_

Round 1

Reviewer 1 Report

It is quite valuable to explore the role of associations in reducing the emotional and financial impact on relatives of caring for children with Duchenne. The authors demonstrate the differences between the two countries through their study, which is an extension of research in the field.

1  Based on the entire manuscript, the authors compared the impact of different health care models in the two countries on the carer for children with Duchenne, rather than the impact of specific associations or organizations. The title and description in the text should be more precise and clear.

2 The Introduction section needs revision. There are many paragraphs at present, and the content description is a little confusing. It is recommended that the author elaborate on the background , the study purpose , and the current study status (although the author talks about there are no previous studies analyzing the positive impact of associations on the lives of parents with children with DMD. However, the manuscript mentions a large number of related research), research problems to be solved, etc. In particular, the literature as indicators of emotional and financial impact should be supplemented.

3 In the measurement part, each variable needs to clearly write the operational definition, measurement method, source of the scale, number of items, examples of items, etc.

4 For all measures used, please indicate whether response scores were summed or averaged to create their composite scores for data analysis.

5 Overall, the discussion sectionis a little brief and underdeveloped. I suggest the authors review this section and make an effort to describe the difference between the two countries' carers and the possible reasons for the difference item by item .

6 The practical implications of the research can be analyzed from the aspects of government, society, organization, individual and so on. Finding differences between countries in cross-cultural research is not the goal; research needs to provide valuable suggestions for improving practice.

7 .Please, revise reference according to APA style. Some Doi are missing.

Author Response

Response to Reviewer

     First of all, I would like to thank the reviewer for it helpful suggestions. I am sure that these changes will contribute to the improvement of the paper. The suggested revisions are described below mentioning point-by-point what changes have been made. After each suggestion, the answer is described written in bold.

Reviewer#1:

Point 1. Based on the entire manuscript, the authors compared the impact of different health care models in the two countries on the carer for children with Duchenne, rather than the impact of specific associations or organizations. The title and description in the text should be more precise and clear.

Response 1: In the introduction text, a sentence has been included that justifies the role of associations as a subsidiary mechanism to health systems, and therefore the objective is to analyse the role played by associations (Line number 40-41). Health systems may vary between Spain and Mexico, and depending on this, associations will play one role or another, but our aim has been to analyse the role of associations as complementary to health systems. The introduction has been rewritten to be more specific and clearer about the objective of the study, as well as the discussion.

Point 2. The Introduction section needs revision. There are many paragraphs at present, and the content description is a little confusing. It is recommended that the author elaborate on the background , the study purpose , and the current study status (although the author talks about there are no previous studies analyzing the positive impact of associations on the lives of parents with children with DMD. However, the manuscript mentions a large number of related research), research problems to be solved, etc. In particular, the literature as indicators of emotional and financial impact should be supplemented.

Response 2: The introduction has been thoroughly revised and edited. The objective and current status of the topic have been established (131-135). Gaps in research, and the reasons why such a study is important, have been added. (Line number: 124-130).

Point 3. In the measurement part, each variable needs to clearly write the operational definition, measurement method, source of the scale, number of items, examples of items, etc.

Response 3: The measurement part has been further developed, it has been added the operational definition, measurement method, source of the scale, number of items, and example of items in each variable (Line number: 166-168; 173-177; 184; 190).

Point 4. For all measures used, please indicate whether response scores were summed or averaged to create their composite scores for data analysis.

Response 4: It has been added how composite scores for data analysis were created. In relation to this point, paragraphs have been added in the statistical section (Line number: 228-233).

Point 5. Overall, the “discussion section” is a little brief and underdeveloped. I suggest the authors review this section and make an effort to describe the difference between the two countries' carers and the possible reasons for the difference item by item. 

Response 5: The entire discussion section has been rewritten and developed, organizing it, and generating important conclusions and further discussion of all the results obtained.

Point 6. The practical implications of the research can be analyzed from the aspects of government, society, organization, individual and so on. Finding differences between countries in cross-cultural research is not the goal; research needs to provide valuable suggestions for improving practice (845-850).

Response 6: It has been added practical implications and the conclusion section has been rewritten also, in order to improve this and to develop valuable suggestions. (Line number: 462-469; 471-487).

Point 7. Please, revise reference according to APA style. Some Doi are missing.

Response 7: References according to the style that is required have been revised. Some DOI that were missing have been added.

Reviewer 2 Report

In this study, the authors examined the influence of associations (NGOs) on various aspects of the life of informal carers (relatives) of a child with Duchenne Muscular in two countries, Mexico and Spain. 74 informal carers filled out verified questionnaires. The analysis of the questionnaire showed that there are differences in the work of the associations between these two countries and that the caregivers in Mexico think that the associations help them more than the caregivers in Spain think. The financial burden on caregivers in both countries is relatively high, with costs including patient and carer health services, as well as the cost of assistive technology. However, the physical symptoms reported by caregivers from Spain and their life satisfaction are worse and the overload is higher than that of caregivers from Mexico, which may be partly attributable to cross-cultural differences.

The authors are aware that the study is biased by the fact that the sample from Mexico consists only of women. I wonder why they didn't compare the results of questionnaires only between women from Mexico and Spain (excluding the 10 men)? Another thing that is not clear to me is the consistent use of the term "relatives": do the sample of carers consist only of mothers and fathers of sick children or other relatives as well? Because their own involvement in the care of sick children could be perceived differently by their brothers and sisters (or other relatives) than by their parents.

I strongly recommend that authors have their manuscript read by a native speaker; I started to correct the manuscript and restructure the sentences to be in English language style, but I gave up after page 3.

After the necessary corrections, I recommend the manuscript for publication.

Remarks:

line 3 (title) – please delete „of“

line 4 – please start the word „muscular“ with small letter

lines 5-6 – please write full names of co-authors

lines 14-15 – I believe that only the phone number and e-mail address of the corresponding author is sufficient, because everything else is already written in the affiliation of the first author

Abstract

line 17 – please change „Carers` emotions and finances“ to „The emotions and finances of carers …“; change „deterioration in“ to „deterioration of“

line 20 – please add „the quality“

line 21 – please change „was composed“ to „consisted“

line 22 – please delete the article „a“

lines 22-23 – the authors use a hyphen for the word „socio-economic“, but not for „sociodemographic“, please make it uniform

line 24 – please delete „the“ in front of „carers“ (two times in this line), please change „had“ to „were in“

lines 25-28 – the font size is smaller than in the rest of the abstract

line 25 – please change „part amount“ to „proportion“

line 26 – please change „professionals“ to „services“

line 27 – please delete „the“ in front of „support“; change „an essential“ to „a key“

Keywords – please add „non-governmental organizations“, „Mexico“ and „Spain“

line 33 – please change this sentence to „Over the years, interest in and awareness of rare diseases (RDs) have grown in both the state and society.“

line 35 – please change „chronic disease“ to „chronic illness on patients` lives“

line 36 – please delete „however, “

line 37 – please change „patients with RDs have been“ to „RD patients are only“

line 38 – the authors wrote „In policy terms, these diseases have been associated with catastrophic diseases, … “ – It is not clear what you wanted to say here, whether that the political regulations refer to catastrophic diseases and other mentioned diseases?

lines 42-43 – please rephrase to „In this case, national plans, programs and strategies that deal with the needs of this group are missing.“

line 45 – please delete the first sentence

lines 45-46 – please rephrase to „In Spain, a national strategy was presented back in 2009 in order to adapt healthcare actions to the needs of patients and to better monitor their implementation [6].“

lines 47-48 – please rephrase to „The monitoring showed that healthcare and social systems in Spain do not adequately meet the needs of rare disease patients, which is similar to that in Mexico [7-8].“

lines 49-50 – „Due to the lack of public funds …“; „in the income of patients

line 51 – please change „drugs“ to „medicines“

lines 52-53 – please rephrase to „This can lead to problems in the family budget, which is further aggravated … or quit their jobs due to illness.“

line 56 – change „to RD patients“ to „for RD patients“, change „which reduce“ to „thereby reducing“

line 57 - change „These“ to „They“

line 60 – change „in making them visible“ to „for their visibility; add „… and are considered…“

line 64 – „ … led to the establishment of a network that, among other goals, aims to spread knowledge among the public to improve …“

line 66 – delete „have“; change „an essential“ to „a key“

line 67 – „as they proposed a joint strategy and series“

line 68 – change „are“ to „were“

line 69 – „Some associations in Spain and Mexico give visibility and support“

lines 70-73 – please combine these two sentences into one („… hereditary disorder [20-23] in which progressive muscle degeneration …“)

lines 76-77 – please combine these two sentences into one: „Given that this type of care requires specific expertise, takes a long time, and is associated with additional financial costs, its impact on the carer's life is very noticeable.“

line 78 – „… quit their jobs“

line 79 – „who continue to work“; „productivity declines“; change „high levels of absenteeism“ to „frequent absences from work“

line 84 – „… deterioration of …“

line 85 – change „The care given to an individual with“ to „Caring for a person with …“

lines 89-90 – „… suffer a significant overload, which makes their social life difficult [29].“ (I am not sure about the second part of this sentence, is that what you meant?)

line 92 – change „On the other hand,“ to „In addition,“; „life satisfaction“ to „satisfaction with life“

line 93 – „is also worse than that of parents …“

line 94 – „Due to the impact …“

lines 95-96 – „… previous studies that analyze the …“; „… parents of children …“

line 97 – „particularly worth analyzing two models of health care.

line 98 – change „and therefore“ to „so“

line 100 –„parents are generally not scheduled for all health examinations on the same day“

line 101 - please change „assume“ to „take over“

line 102 – change „On the contrary“ to „Conversely“

line 103 – change „assume the cost of the care“ to „cover the cost of care“

line 104 – „who all see the patient on the same day.“

line 105 – „also to other“

line 108 – „Because of all the above, …“

line 109 – „quality of life of caregivers“

line 110 – please change „resulting from the disease“ to „arising from illness“

line 116 – please change „resided in“ to „from“; delete „and“

line 118 - please change „resided in“ to „from“; delete „and“

lines 121-121 – please delete „an informal carer“ in bullet b) (you are repeating yourself)

line 125 – please add bold words „ Exclusion criteria: a) informal carer of a child with any other …“

line 136 – please rephrase to „… questionnaire was used to assess the ...“

line 137 – „It includes the questions on the assistive technology that the family had to buy, the money …“

line 138 – „… the time …“

line 139 – these is an extra closing parenthesis before the period

Table 2 - the title is not clear, I suggest something like „Percentage of annual income related to carers` health expenses“. Please change the sign (€) in the table header to (%), and consequently delete the „%“ sign from all other cells.

line 248 – you cite yourself here using „(Aranillas et al., 2010)“ instead of [50]

Table 3 – please indicate in some way the variables that are significantly different (let them be bold or in italics)

line 266 – „in reducing the emotional and economic impact of illness on carers,

line 387 – correct „MS and JFLP“

line 396 – delete „Informed Consent Statement“

Author Response

Response to Reviewer

     First of all, I would like to thank the reviewer for it helpful suggestions. I am sure that these changes will contribute to the improvement of the paper. The suggested revisions are described below mentioning point-by-point what changes have been made. After each suggestion, the answer is described written in bold.

Reviewer#2:

In this study, the authors examined the influence of associations (NGOs) on various aspects of the life of informal carers (relatives) of a child with Duchenne Muscular in two countries, Mexico and Spain. 74 informal carers filled out verified questionnaires. The analysis of the questionnaire showed that there are differences in the work of the associations between these two countries and that the caregivers in Mexico think that the associations help them more than the caregivers in Spain think. The financial burden on caregivers in both countries is relatively high, with costs including patient and carer health services, as well as the cost of assistive technology. However, the physical symptoms reported by caregivers from Spain and their life satisfaction are worse and the overload is higher than that of caregivers from Mexico, which may be partly attributable to cross-cultural differences.

Point 1. The authors are aware that the study is biased by the fact that the sample from Mexico consists only of women. I wonder why they didn't compare the results of questionnaires only between women from Mexico and Spain (excluding the 10 men)?

Response 1: Thank you for your consideration. The exclusion of 10 men would significantly affect the quality of the results. Given that we found no statistical intra-group and inter-group gender differences, it was considered not to exclude them from the study. In some cases, the main caregiver is male, depending on the family structure, and therefore, given that he is the main caregiver, the impact on his psychological well-being should not be omitted. On the other hand, this study mainly compares two systems of patient organisations and the services they provide to primary caregivers irrespective of their gender.

Point 2. Another thing that is not clear to me is the consistent use of the term "relatives": do the sample of carers consist only of mothers and fathers of sick children or other relatives as well? Because their own involvement in the care of sick children could be perceived differently by their brothers and sisters (or other relatives) than by their parents.

Response 2: The sample consist of mothers and fathers only. It has been eliminated the word „relatives“ throughout the text in order to avoid misunderstandings. In addition, a limitation of the study was the fact that it was not possible to include other caregivers, such as siblings, aunts, ancles, etc. This limitation has been added (Line number: 450-452).

Point 3. I strongly recommend that authors have their manuscript read by a native speaker; I started to correct the manuscript and restructure the sentences to be in English language style, but I gave up after page 3.

Response 3: As we are not English native speakers, we have thoroughly edited the document to ensure that no language issues remain in relation to the content of the paper.

Point 4. Remarks:

line 3 (title) – please delete „of“

This change has been made.

line 4 – please start the word „muscular“ with small letter

This change has been made.

lines 5-6 – please write full names of co-authors

Full names o co-author have been added.

lines 14-15 – I believe that only the phone number and e-mail address of the corresponding author is sufficient, because everything else is already written in the affiliation of the first author

It has been eliminated the address of the corresponding author because is already written in the affiliation.

Abstract

line 17 – please change „Carers` emotions and finances“ to „The emotions and finances of carers …“; change „deterioration in“ to „deterioration of“

This change has been made.

line 20 – please add „the quality“

This change has been made.

line 21 – please change „was composed“ to „consisted“

This change has been made.

line 22 – please delete the article „a“

This change has been made.

lines 22-23 – the authors use a hyphen for the word „socio-economic“, but not for „sociodemographic“, please make it uniform

These words have been made uniform.

line 24 – please delete „the“ in front of „carers“ (two times in this line), please change „had“ to „were in“

This change has been made.

lines 25-28 – the font size is smaller than in the rest of the abstract

The font size have been changed.

line 25 – please change „part amount“ to „proportion“

This change has been made.

line 26 – please change „professionals“ to „services“

This change has been made.

line 27 – please delete „the“ in front of „support“; change „an essential“ to „a key“

This change has been made.

Keywords – please add „non-governmental organizations“, „Mexico“ and „Spain“

These key words have been added.

line 33 – please change this sentence to „Over the years, interest in and awareness of rare diseases (RDs) have grown in both the state and society.“

This change has been made.

line 35 – please change „chronic disease“ to „chronic illness on patients` lives“

This change has been made.

line 36 – please delete „however, “

This change has been made.

line 37 – please change „patients with RDs have been“ to „RD patients are only“

This change has been made.

line 38 – the authors wrote „In policy terms, these diseases have been associated with catastrophic diseases, … “ – It is not clear what you wanted to say here, whether that the political regulations refer to catastrophic diseases and other mentioned diseases?

We want to explain that authorities haven’t defined correctly this type of diseases. This sentence has been explained in order to avoid misunderstanding.

lines 42-43 – please rephrase to „In this case, national plans, programs and strategies that deal with the needs of this group are missing.“

It has been rephrased.

line 45 – please delete the first sentence

It has been deleted.

lines 45-46 – please rephrase to „In Spain, a national strategy was presented back in 2009 in order to adapt healthcare actions to the needs of patients and to better monitor their implementation [6].“

It has been rephrased

lines 47-48 – please rephrase to „The monitoring showed that healthcare and social systems in Spain do not adequately meet the needs of rare disease patients, which is similar to that in Mexico [7-8].“

It has been rephrased.

lines 49-50 – „Due to the lack of public funds …“; „in the income of patients

This change has been made.

line 51 – please change „drugs“ to „medicines“

This change has been made.

lines 52-53 – please rephrase to „This can lead to problems in the family budget, which is further aggravated … or quit their jobs due to illness.“

It has been rephrased.

line 56 – change „to RD patients“ to „for RD patients“, change „which reduce“ to „thereby reducing“

This change has been made.

line 57 - change „These“ to „They“

This change has been made.

line 60 – change „in making them visible“ to „for their visibility; add „… and are considered…“

This change has been made.

line 64 – „ … led to the establishment of a network that, among other goals, aims to spread knowledge among the public to improve …“

This change has been made.

line 66 – delete „have“; change „an essential“ to „a key“

It has been deleted.

line 67 – „as they proposed a joint strategy and series“

This change has been made.

line 68 – change „are“ to „were“

This change has been made.

line 69 – „Some associations in Spain and Mexico give visibility and support“

This change has been made.

lines 70-73 – please combine these two sentences into one („… hereditary disorder [20-23] in which progressive muscle degeneration …“)

This change has been made.

lines 76-77 – please combine these two sentences into one: „Given that this type of care requires specific expertise, takes a long time, and is associated with additional financial costs, its impact on the carer's life is very noticeable.“

These two sentences have been combined.

line 78 – „… quit their jobs“

This change has been made.

line 79 – „who continue to work“; „productivity declines“; change „high levels of absenteeism“ to „frequent absences from work“

This change has been made.

line 84 – „… deterioration of …“

This change has been made.

line 85 – change „The care given to an individual with“ to „Caring for a person with …“

This change has been made.

lines 89-90 – „… suffer a significant overload, which makes their social life difficult [29].“ (I am not sure about the second part of this sentence, is that what you meant?)

Yes, it was what we mean. We have corrected it.

line 92 – change „On the other hand,“ to „In addition,“; „life satisfaction“ to „satisfaction with life“

This change has been made.

line 93 – „is also worse than that of parents …“

This change has been made.

line 94 – „Due to the impact …“

This change has been made.

lines 95-96 – „… previous studies that analyze the …“; „… parents of children …“

This change has been made.

line 97 – „particularly worth analyzing two models of health care.

This change has been made.

line 98 – change „and therefore“ to „so“

This change has been made.

line 100 –„parents are generally not scheduled for all health examinations on the same day“

This change has been made.

line 101 - please change „assume“ to „take over“

This change has been made.

line 102 – change „On the contrary“ to „Conversely“

This change has been made.

line 103 – change „assume the cost of the care“ to „cover the cost of care“

This change has been made.

line 104 – „who all see the patient on the same day.“

This change has been made.

line 105 – „also to other“

This change has been made.

line 108 – „Because of all the above, …“

This change has been made.

line 109 – „quality of life of caregivers“

This change has been made.

line 110 – please change „resulting from the disease“ to „arising from illness“

This change has been made.

line 116 – please change „resided in“ to „from“; delete „and“

This change has been made.

line 118 - please change „resided in“ to „from“; delete „and“

This change has been made.

lines 121-121 – please delete „an informal carer“ in bullet b) (you are repeating yourself)

This change has been made.

line 125 – please add bold words „ Exclusion criteria: a) informal carer of a child with any other …“

This change has been made.

line 136 – please rephrase to „… questionnaire was used to assess the ...“

This change has been made.

line 137 – „It includes the questions on the assistive technology that the family had to buy, themoney …“

This change has been made.

line 138 – „… the time …“

This change has been made.

line 139 – these is an extra closing parenthesis before the period

This change has been made.

Table 2 - the title is not clear, I suggest something like „Percentage of annual income related to carers` health expenses“. Please change the sign (€) in the table header to (%), and consequently delete the „%“ sign from all other cells.

This change has been made.

line 248 – you cite yourself here using „(Aranillas et al., 2010)“ instead of [50]

This change has been made.

Table 3 – please indicate in some way the variables that are significantly different (let them be bold or in italics)

This change has been made.

line 266 – „in reducing the emotional and economic impact of illness on carers,

This change has been made.

line 387 – correct „MS and JFLP“

This change has been made.

line 396 – delete „Informed Consent Statement“

This change has been made.

Reviewer 3 Report

1.      The title should remain as follows: The role of associations in reducing the emocional  emotional and financial impact on relatives of caring for children with Duchenne Muscular dystrophy: a cross-cultural study.

2.      In the introduction, please also specify that: The prevalence of DMD is less than 10 cases per 100,000 males and seems to be the same between regions, and DMD in females is very rare (<1 per million) and is limited to case reports of individuals with Turner syndrome, a translocation involving DMD or those with bi-allelic DMD mutations (Duan, D., Goemans, N., Takeda, S. et al. Duchenne muscular dystrophy. Nat Rev Dis Primers 7, 13 (2021). https://doi.org/10.1038/s41572-021-00248-3)

3.      The introduction is very well grounded, it contains a lot of bibliographical sources, but still, I did not find anywhere the novative/innovative/original elements of the study. As such, please insert the elements of originality at the end of the introduction.

4.      I understand that the present study aims to analyze the role of different associations in improving the quality of life for caregivers of children with DMD. Please highlight the purpose of this article and relate it to the elements of originality.

5.      A 248-item questionnaire assessing the financial costs associated with care was used (line 136). This questionnaire is supported by studies conducted with people with NMD (neuromuscular diseases). Here we will need some studies conducted on people with NMD or DMD [26 and].

6.      PHQ-15 [34] (line 143) must remain PHQ-15 [34,26]

7.      Nowhere in the paper did I find the sample size or statistical power calculation. Please enter the sample size calculation.

8.      Please rethink the conclusions part, without any citation. Conclusions should be concise and to the point.

9.      The paper has very low similarity coefficients, but still, one paragraph needs to be reworded entirely:

·        The values of the CarerQol-7D ranged between 0.59 and 0.81. Intraclass Correlation Coefficients (ICCs) of the CarerQol7D had values between 0.55 and 0.94 [45].

Author Response

Response to Reviewers

     First of all, I would like to thank the reviewer for its helpful suggestions. I am sure that these changes will contribute to the improvement of the paper. The suggested revisions are described below mentioning point-by-point what changes have been made. After each suggestion, the answer is described written in bold.

Reviewer#3:

Point 1. The title should remain as follows: The role of associations in reducing the emocional  emotional and financial impact on relatives of caring for children with Duchenne Muscular dystrophy: a cross-cultural study.

Response 1: The title has been rewritten (Line number: 2-4).

Point 2. In the introduction, please also specify that: The prevalence of DMD is less than 10 cases per 100,000 males and seems to be the same between regions, and DMD in females is very rare (<1 per million) and is limited to case reports of individuals with Turner syndrome, a translocation involving DMD or those with bi-allelic DMD mutations (Duan, D., Goemans, N., Takeda, S. et al. Duchenne muscular dystrophy. Nat Rev Dis Primers 7, 13 (2021). https://doi.org/10.1038/s41572-021-00248-3)

Response 2: This sentence has been added (Line number: 84-87)

Point 3. The introduction is very well grounded, it contains a lot of bibliographical sources, but still, I did not find anywhere the novative/innovative/original elements of the study. As such, please insert the elements of originality at the end of the introduction.

Response 3: The elements of originality at the end of the introduction have been added (Line number: 40-43; 124-130; 133-135).

Point 4. I understand that the present study aims to analyze the role of different associations in improving the quality of life for caregivers of children with DMD. Please highlight the purpose of this article and relate it to the elements of originality.

Response 4: The purpose of this article related to the elements of originality in this study has been added (Line number: 133-135).

Point 5. A 248-item questionnaire assessing the financial costs associated with care was used (line 136). This questionnaire is supported by studies conducted with people with NMD (neuromuscular diseases). Here we will need some studies conducted on people with NMD or DMD [26 and].

Response 5: There has been added studies conducted with people with NMD by which this questionnaire was supported (Line number: 168-169). A new sentence that explains this has been added.

Point 6. PHQ-15 [34] (line 143) must remain PHQ-15 [34,26]

Response 6: This section only mentions studies of the design and validation of the original test [34], as well as the adaptation and validation with a Spanish sample [35].

Point 7. Nowhere in the paper did I find the sample size or statistical power calculation. Please enter the sample size calculation.

Response 7: There are no epidemiological data in Spain and Mexico on the number of cases of patients diagnosed with DMD. It would only be possible to establish an estimate based on data from other countries. For this reason, it has not been included. There are no official data published by the state or in epidemiological journals.

Point 8. Please rethink the conclusions part, without any citation. Conclusions should be concise and to the point.

Response 8: The conclusions part has been rewritten in order to be concise and to generate important conclusions (471-487).

Point 9. The paper has very low similarity coefficients, but still, one paragraph needs to be reworded entirely: The values of the CarerQol-7D ranged between 0.59 and 0.81. Intraclass Correlation Coefficients (ICCs) of the CarerQol7D had values between 0.55 and 0.94 [45].

Response 9: This paragraph has been rewritten (Line number: 204-205).

Reviewer 4 Report

The Authors present a paper:" The role of associations in reducing the emotional and financial impact of caring for children with Duchenne Muscular dystrophy: a cross-cultural study" interesting as content but with a poor general and widespread diffusion. The study, well done, is related to  specific countries situation and is difficult to generalize for other ethnical population having different socioeconomic status. The aim of their study is to analyze the positive impact of associations on the lives of parents with  children with DMD looking at two health care models in Spain and in Mexico but what is really the teaching from this study? 

Author Response

Response to Reviewers

     First of all, I would like to thank the reviewer for its helpful suggestions. I am sure that these changes will contribute to the improvement of the paper. The suggested revisions are described below mentioning point-by-point what changes have been made. After each suggestion, the answer is described written in bold.

Reviewer#4:

Point 1. The Authors present a paper:" The role of associations in reducing the emotional and financial impact of caring for children with Duchenne Muscular dystrophy: a cross-cultural study" interesting as content but with a poor general and widespread diffusion. The study, well done, is related to  specific countries situation and is difficult to generalize for other ethnical population having different socioeconomic status. The aim of their study is to analyze the positive impact of associations on the lives of parents with  children with DMD looking at two health care models in Spain and in Mexico but what is really the teaching from this study? 

Response 1: This study shows that the pattern of care provided from one association to another, depending on culture, has a more positive effect on the well-being of caregivers. However, the positive effect attributable to Mexican associations is due to the fact that these organisations take on the competences of the state. Paradoxically, in the case of Spain, caregivers have a lower level of well-being, although they receive more medical services free of charge, albeit in a fragmented and uncoordinated way among health professionals. Therefore, we conclude that the way care is received has an influence and that associations assume roles that do not correspond to them. Practical implications arising from the results of this study have been added, as well as new paragraphs throughout the discussion explaining the importance of carrying out these studies. The conclusions have also been edited (471-486), and a paragraph that explain the practical implications of the study has also been added. (Line number: 461-468).

Round 2

Reviewer 1 Report

I think the authors have responded to my concerns and made proper revisions. The quality of the manuscript has been significantly improved for publication.